# Two major ecological shifts shaped 60 million years of ungulate faunal evolution

Fernando Blanco [1,2,3] ✉, Ignacio A. Lazagabaster [3,4,5], Óscar Sanisidro [6], Faysal Bibi [3], Nicola S. Heckeberg [7,8], María Ríos [9,10], Bastien Mennecart [11], María Teresa Alberdi [12], Jose Luis Prado [13], Juha Saarinen [14], Daniele Silvestro [1,2,15], Johannes Müller [3], Joaquín Calatayud [16,17] & Juan L. Cantalapiedra [3,6,12]

The fossil record provides direct evidence for the behavior of biological systems over millions of years, offering a vital source for studying how ecosystems evolved and responded to major environmental changes. Using network analysis on a dataset of over 3000 fossil species spanning the past 60 Myr, we find that ungulate continental assemblages exhibit prolonged ecological stability interrupted by irreversible reorganizations associated with abiotic events. During the early Cenozoic, continental assemblages are dominated by mid-sized browsers with low-crowned teeth, which show increasing functional diversity. Around 21 Ma, the formation of a land bridge between Eurasia and Africa triggers the first major global transition towards a new functional system featuring a prevalence of large browsers with mid- to high-crowned molars. Functional diversity continues to increase, peaking around 10 Ma. Shortly after, aridification and the spread of C4-dominated vegetation lead to a second tipping point towards a fauna characterized by grazers and browsers with high and low crowned teeth. A global decline in ungulate functional diversity begins 10 Ma ago and accelerates around 2.5 Ma, yet the functional structure of these faunas remains stable in the latest Cenozoic. Large mammal evolutionary history reflects two key transitions, aligning with major tectonic and climatic events.

The ongoing biotic crisis is characterized by a rapid deterioration of most Earth's ecosystems[1], which could be irrevocably tipping several terrestrial and marine ecosystems towards new ecological regimes (i.e. functional tipping points[2–4]). Such shifts in functional states, which are abrupt and irreversible even over geological time, have been identified in the fossil record[5,6], suggesting that past perturbations can be used as analogs to gain insights about the tempo and nature of immediate and future responses to environmental changes[7–9]. Faunal shifts in the fossil record have been traditionally tackled from a taxonomic perspective (i.e., changes in the taxonomic composition of communities and faunas[10,11]). More recently, paleobiological studies have progressively incorporated taxon-free functional approaches, utilizing

changes in ecomorphological composition to evaluate ecological shifts beyond taxonomic turnover[5,12–14].

In terrestrial ecosystems, most of these functional approaches have focused on patterns of collapse of mammalian assemblages during the late Quaternary[12–14]. A deeper temporal dimension of taxon–free approaches is needed to provide a more profound understanding of the evolution of mammalian ecosystem dynamics and how they responded to major environmental disruptions. Broadening the temporal scope should also reveal the mechanisms related to other patterns beyond collapse episodes, like those involved in modulating ecological resilience and stasis[15], as well as processes underpinning dynamics of recovery and the unfolding of functional diversity.

Because environmental change has a huge impact on ecosystem primary productivity and biomass transfer[16–18], investigating long-term patterns in primary consumer guilds should provide valuable insights into major ecological reconfigurations. Mammalian large herbivores have been the major vertebrate primary consumers in our planet's ecosystems in Cenozoic times. Ungulates consume disproportionately more plant biomass per unit area than any other vertebrate group[19]. Many large mammalian herbivores are considered ecosystem engineers because they play a crucial role in processes involved in fire regimes control, seed dispersal, ecological succession of ecosystems, and soil compaction[19–25]. Crucially, because of the recent and ongoing decline of large herbivores, these functions are largely missing from modern ecosystems[24]. In a search for a broader perspective on the behavior of terrestrial functional systems, we ask: how did terrestrial functional faunas shift throughout the Cenozoic? Are functional tipping points a common feature in these systems? If so, which factors triggered such state transitions?

Here, we aim to identify the main shifts in global large herbivore functional structure of the orders Artiodactyla, Perissodactyla, and Proboscidea, which constitute most of the ungulate fossil record of the Cenozoic. As ecosystem engineers, terrestrial ungulates have a tight connection to physical changes and habitat disruptions, making them ideal to study current and past ecosystem functional transitions[19,25]. We developed an extensive database for large herbivores, containing occurrence information for 3012 species spanning the past 60 million years. For each species, we gathered information for 14 ecomorphological traits (13 dental traits and body size) and clustered species based on their pairwise functional distances. We refer to these groups as 'functional types' (FT), since their constituent species would fulfill comparable ecological roles. We then applied Network Analysis to identify ungulate continental functional faunas (UFFs)[5], defined as unique associations of functional types. Notably, UFFs are not defined solely by the presence of FTs, but by their relevance, affinity, and fidelity in the faunas (see methods). Our network analysis ignores assemblage age (it focuses exclusively on the functional structure), and the temporal succession of UFFs is determined *a posteriori* (see methods). Lastly, we compared the results of our functional analyses with taxonomic faunas and the timing of major environmental shifts of the Cenozoic[26–34]. Our results unveil two major shifts in the UFF succession coinciding with abiotic events.

## Results and discussion

Our results show a decoupling of taxonomic and functional assembly dynamics (Fig. 1A and Supplementary Fig. 1). The functional structure of large herbivore assemblages exhibits greater resilience compared to their taxonomic structure. On average, taxonomic reassembly (module change, see Supplementary Fig. 1) occurred every $1.7 \pm 0.31$ Myr, whereas functional reassembly took place every $6.25 \pm 1.6$ Myr. Sensitivity analyses validate these findings, ruling out potential artifacts stemming from sampling, aggregation or the selection of community detection algorithms (see methods). Large herbivore assemblages experienced long periods of ecological stasis, punctuated by two episodes of global and almost synchronous functional reassembly (tipping points, Fig. 1A). These tipping points occurred around 21 and 10 Ma, likely as a response to major abiotic events. Since 10 Ma, large herbivore assemblages have exhibited remarkable stability, first in Africa, followed by the Americas and Europe around 7 Ma, and later in Asia around 4.5 Ma (Fig. 1A).

### Early Cenozoic stability

For most of the Cenozoic, the faunas of America, Europe, and Asia were grouped into a single module, representing a shared functional configuration (Fig. 1A), and occupied similar regions of functional space (Fig. 1B). Before 30 Ma, African faunas included only proboscideans among the orders examined, preventing a comprehensive

assessment of the continent's functional structure during this time period.

The Paleogene was a time dominated by mid-size browsers with low-crowned teeth (brachydont) (Supplementary Figs. S4 and S17). Between 60 and 40 Ma, the system experienced a long period of functional stability (ecological stasis), with a first reconfiguration in UFFs around 40 Ma, when American ungulate assemblages diverged functionally from those in Europe and Asia (Fig. 1A).

During the early Cenozoic, there is a global increase in functional diversity (Fig. 2). Functional diversity encapsulates both functional richness (the number of functional types) and functional evenness (the degree to which species are equally divided across functional types, see methods). Initially, this increase in functional diversity is primarily attributable to an increase in functional richness (Fig. 2), especially in American faunas, which exhibited a sustained increase in functional diversity between 50 and 20 Ma (Fig. 2B). Functional innovations were likely intertwined with biogeographic events, contributing to this enrichment. For example, around 37.5 Ma, a significant immigration of large mammals from Asia to North America occurred following a sea-level drop[10]. In contrast, ungulate faunas of Europe and particularly Asia underwent a prolonged functional diversity decline during the early Cenozoic (Fig. 2). At about 40 Ma, these diverging trends culminated in the split of American ungulate assemblages from those in Europe and Asia (Fig. 1A, B). This was primarily driven by the emergence of functional dominance of FTs in larger body size categories in the Americas, in contrast to the mid-sized herbivores that characterize the UFF of Eurasia (Supplementary Fig. 17). The geographical extent of landmasses could have significantly influenced the survival dynamics of different faunal groups. Specifically, the broader southern reach of the Eurasian landmass compared to North America may have provided a more stable environment for mid-sized herbivores. This stability persisted despite the overarching trend of Cenozoic cooling, which likely affected species differently depending on their latitudinal distribution.

Europe and Asia formed a single UFF and shared a common functional trajectory until around ca. 32 Ma, when the European UFF diverged from the shared module with Asia (Fig. 1A). This divergence coincided with a period of global cooling that ultimately led to the formation of permanent ice sheets in Antarctica, marking the transition from the greenhouse climate of the Early Paleogene to the onset of the Late Cenozoic icehouse climate. This climatic shift, known as the Eocene-Oligocene transition (EOC), triggered major faunal extinctions and turnover, primarily in Europe, in a faunal event also dubbed as the Grande Coupure[26,35]. Characterized by the influx of migrant faunas from Asia to Europe, the Grande Coupure replaced archaic ungulates with modern faunas[35,36]. The asymmetry in the replacement is explained by differences in *tempo*, biogeography and differential climatic regimes[26,37]. The EOC caused less faunal disruption in Asia, where archaic forms survived until the late Early Oligocene, retaining a faunal assemblage more similar to the ones of the Late Eocene (e.g., archaic Amynodontidae forms, such as *Amynodon* or *Sianodon*)[37]. Furthermore, modern forms gradually appeared in Asia prior to the EOC, including some families that had already emerged in the late Eocene (e.g., Enteledontidae), in contrast to the abrupt replacement observed in European faunas immediately after the EOC[37]. This abrupt faunal replacement drove a decline in the functional diversity of European faunas, primarily due to a reduction in functional richness (Fig. 2). This, combined with the abrupt loss of archaic ungulates and their replacement by modern forms, caused the European faunal assemblages to diverge from the late Eocene ones, which remained an important component of Asian faunas. The biogeographic and climatic context of the EOC transition also played a crucial role in shaping the observed functional shifts[26]. Europe's restricted high-latitude position and its configuration as an archipelago during this period may have constrained faunal dispersal and climate tracking, making species more

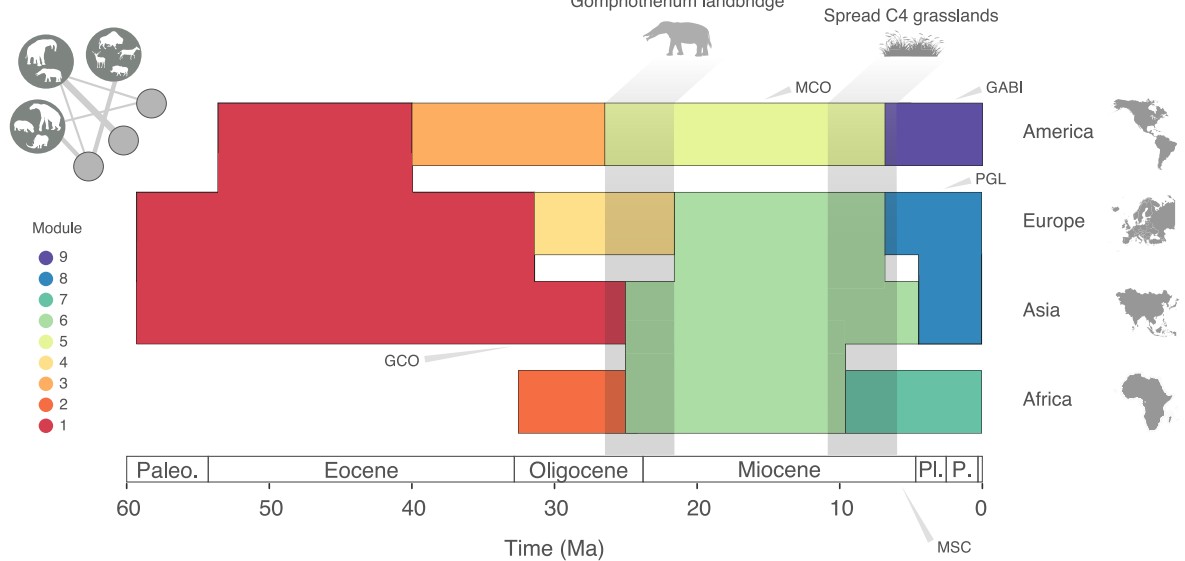

**A**  Functional structure dynamics

**B**  Functional space by continent

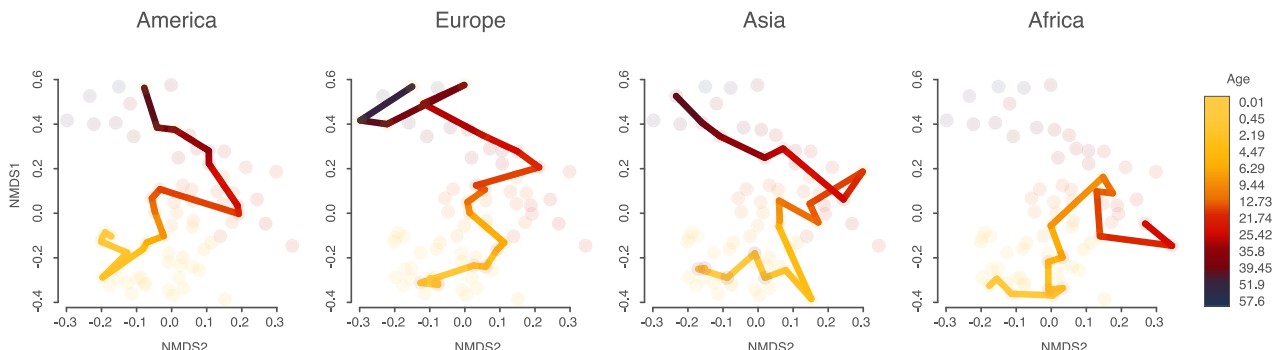

**Fig. 1 | Functional structure dynamics in continental ungulate assemblages.**
**A** The nine main functional modules plotted across time and by continent. Note distinct differences between Paleogene and Neogene modules, and also shared modules across Africa and Eurasia, and the long-term distinctiveness of the American continents. Since the African dataset prior to 34 Ma only includes proboscideans, we decided to exclude it from the analysis. **B** Trajectories of each continent through the functional space plotted using non-metric multidimensional

scaling (NMDS). Dots represent time bins in each continent. The line of best fit shows functional changes across time. Animal silhouettes are from phylopic.org, and map outlines are from freevectormaps.com. Paleo. Paleocene, Pl. Pleiocene, P. Pleistocene, GCO Grande Coupure, MCO Middle Miocene Climatic Optimum, MSC Messinian Salinity Crisis, PGL Plio-Pleistocene Glaciations, GABI Great American Biotic Interchange. All data and code for this figure are available in the Source Data file, Supplementary Code and at refs. 82,83.

vulnerable to extinction as conditions changed. In contrast, larger and more continuous landmasses, such as those in North America and Asia, provided broader climatic gradients and refugia, which facilitated species persistence and functional stability (Fig. 1A). These geographic constraints, coupled with the rapid cooling and heightened seasonality of the EOC transition[26], contribute to explaining the pronounced shifts observed in European faunal assemblages. This ultimately led to the divergence of European faunas in terms of functional structure (Fig. 1).

**The African-Eurasian landbridge**
The first global-scale reconfiguration of UFFs took place between 26 and 21 Ma, revealing a major first tipping point. This period was marked by a homogenization of the functional structure of European, Asian, and African faunas (Fig. 1). The functional coalescence is reflected in the drastic reduction of the functional distance between Africa and Eurasia, beginning around 25 Ma, during the Chantian stage (27.82–23.03 Ma) in Europe, and later followed by the Aquitanian (23.03–20.44 Ma) in Asia (Supplementary Fig. 3). A long process of environmental change, marked by a trend towards global aridification,

began around 26 million years ago with the Alpino-Himalayan orogeny[27–29]. This caused the collision of the African and Eurasian plates, which by 21 Ma resulted in the complete closure of the Tethys Sea, the emergence of the Mediterranean Sea, and the formation of a land connection known as the *Gomphotherium* landbridge[30].

The homogenization of faunas and the reassembly into a coherent, transcontinental UFF was therefore probably the result of the terrestrial connection between Eurasia and Africa at this time (Fig. 1A). This resulted in a net gain in functional diversity, with all three orders (Perissodactyla, Artiodactyla, Proboscidea) present across all three continents (Fig. 2). Notably, proboscideans dispersed out of Africa and spread around the world, triggering an acceleration of ecomorphological evolution within this order[38]. However, the impact of the *Gomphotherium* landbridge worked in both directions. Artiodactyls and perissodactyls, which had dominated the large herbivore fauna in Eurasia since the beginning of the Cenozoic, dispersed into Africa while exploring new regions of the functional space (Fig. 1B). Overall, the onset of the tectonic movements that lead to the formation of the *Gomphotherium* landbridge created new ecological configurations,

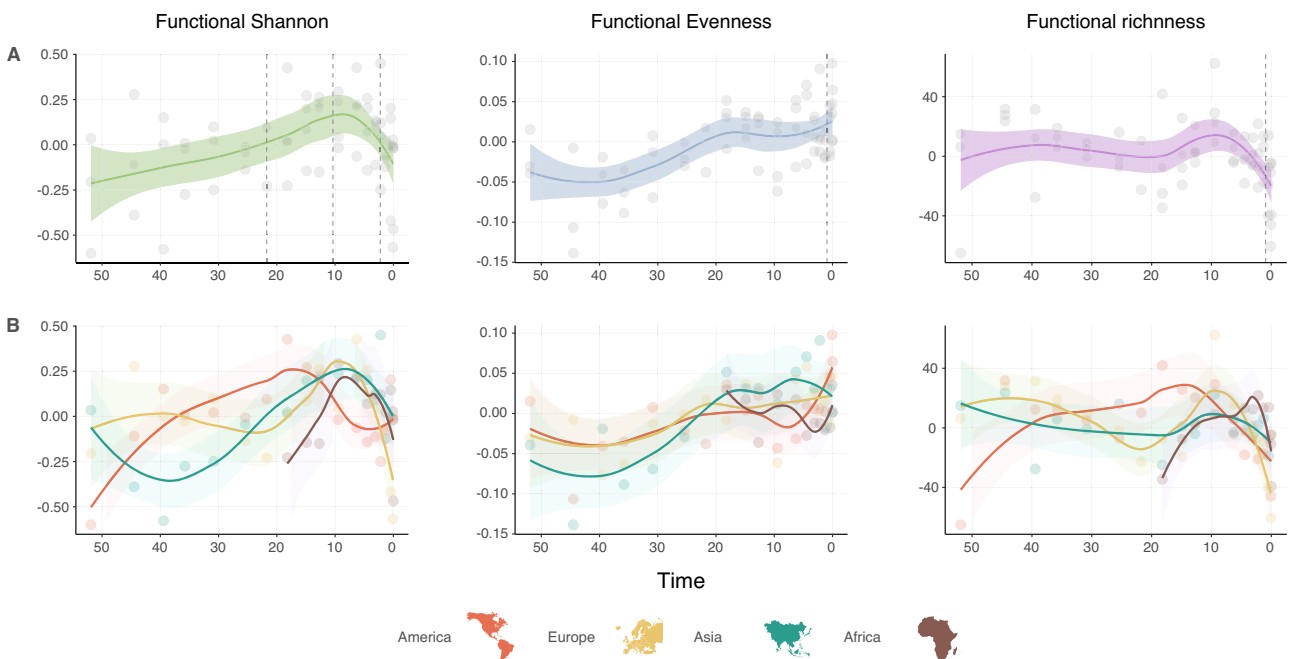

**Fig. 2 | Functional diversity dynamics in continental ungulate assemblages.** Residuals from regressing diversity indexes (Shannon index, evenness index, and functional richness) and number of occurrences are plotted against time in Myr (see methods), showing (**A**) global and (**B**) continental trends. Curves are derived from a local regression fitting (LOESS). The LOESS fit (solid line) represents the smoothed trend, and the shaded areas indicate the 95% confidence interval. Dotted lines are inflection points from regression models with segmented relationships (see methods). Map outlines are from freevectormaps.com. All data and code for this figure are available in the Source Data file, Supplementary Code and at refs. 82,83.

marking a milestone event in the evolutionary history of terrestrial mammal faunas.

Simultaneously, the Americas underwent a reassembly of their functional assemblage, characterized by mid-size grazers with medium to high crowned teeth (Supplementary Figs. 4 and 5). However, the Americas continued to follow their distinct trajectory, retaining an exclusive state in terms of functional structure compared to the other continents (Fig. 1A). In contrast to the trend observed in Eurasia, around 15 Ma the American UFF began to experience a continuous decline in functional diversity (Fig. 2B). Interestingly, this decline follows an immigration event between Eurasia and North America at the end of the early Miocene, some 16.5 Ma[10], suggesting that faunal dispersals do not necessarily enhance functional diversity but may instead reduce the functional richness of endemic American functional types. After 10 Ma, Africa and Eurasia would follow the Americas in this trend of functional diversity decline (Fig. 2B).

### The global expansion of grasslands

Following a global sustained trend of increasing functional diversity fueled by a rise in functional richness (Fig. 2), the functional network reaches a second global tipping point ca. 10 Ma (Fig. 1A)[32,33]. From this moment onwards, we detect a drop in global functional diversity that has persisted to the present day (Fig. 2). This trend resulted in the loss of 65% of functional diversity in less than 5 Myr (from 10 to 5 Ma, see Fig. 2). The functional loss prompted an almost-contemporaneous global reassembly of UFFs characterized by a combination of grazers and browsers with selenodont teeth, represented by ruminants in Africa and Eurasia, and by perissodactyls with bilophodont teeth in the Americas (Supplementary Figs. 4, 5). This reassembly was first observed in African faunas, resulting in their divergence from the previous module shared with Eurasian faunas (Fig. 1). European faunas, followed by the faunas of the Americas and Asia, also underwent a reorganization of their functional structure similar to Africa's (Fig. 1A). This reconfiguration made European and Asian functional assemblages to shortly diverge from each other for two million years (Fig. 1A and

Supplementary Fig. S3). After around 4.5 Ma, the ungulate faunas of both continents would fall back under the same functional configuration, a coherent structure that lasted until current days (Fig. 1A).

Humid, forested environments had been more common prior to the Middle Miocene Climatic Optimum[33], but subsequently the world transitioned towards a drier and colder climate[31]. Increased aridity and declining $CO_2$ atmospheric concentrations favored the expansion of C4-dominated grasslands and drove the development of morphological adaptations to these new environments[39–41]. In this new context, with the need to cope with more abrasive particles in their diet (from plant tissues, soil and grit on the plants), several herbivore lineages developed new dental functional structures and traits, including increased relative tooth height (hypsodonty), larger molar occlusal areas, flatter occlusal topography or increasing number of cutting, transversal lophs, among others[38,42–46]. Consequently, new functionalities emerged as ecological opportunities unfolded, shaping the exploration of different regions within the functional space (Fig. 1B). The novel environmental settings provided ideal conditions for certain groups to diversify (e.g. Equids[47]), while the decline of forest-humid environments worldwide triggered a drastic decrease in global functional diversity[38] (Fig. 2). The decline of the functional diversity that started 10 Ma has continued until the present.

### The quaternary and the functional configuration of modern faunal assemblages

The persistent functional diversity loss that started around 10 Ma was significantly exacerbated around 2.5 Ma, as confirmed by our segmented regression analyses (Fig. 2). Such acceleration in functional disruption was likely due to the severe environmental changes in aridity, vegetation cover, and ecosystem productivity resulting from intensified fluctuations in glacial cycles[31,34,48].

In contrast with other continents, a recovery trend in functional diversity is recorded in the Americas stemming from an increase in functional evenness (Fig. 2A, B). We propose that similar to the formation of the *Gomphotherium* landbridge, the creation of a natural

corridor between North and South America was a key factor driving the rebound in functional diversity. Around 2.6 Ma, the formation of the Panama Isthmus triggered the Great American Biotic Interchange (GABI)[49–54]. The colonization of these new environments by large herbivores from both continents significantly increased their functional diversity, largely through enhanced evenness—species became more evenly distributed across various functional roles within the ecosystems of North and South America (Fig. 2B)[51,55].

However, these changes in functional diversity do not appear to significantly alter the essential functional scaffold of large herbivore continental functional pools (Fig. 1A). From 4.5 Ma, UFFs remained globally unaltered, representing a functional structure stasis that persists to the present day. This stasis has endured even through periods of major abiotic crisis, such as the Messinian Salinity Crisis or the Plio–Pleistocene glaciations, which appear to have affected the functional diversity (Fig. 2). Unlike in past events, the last functional state is resilient enough to withstand the sustained loss of functional diversity while maintaining its essential functional configuration, highlighting the complex interactions between functional structure and diversity (Figs. 1A and 2).

Interestingly, the loss of megaherbivores (>1000 kg) did not entail shifts towards new UFFs (Fig. 1). The decline of these large herbivores began during the later stages of the Cenozoic[38,56]. This functional attrition is better exemplified by the Proboscideans, a once-successful group that reached a global diversity peak of 33 species around 3.2 Ma, just before the intensification of Plio-Pleistocene glaciations (likely a key driver of their decline; 38). The extinction of the largest ungulates accelerated after the Late Pleistocene (129 kya onward), with most species exceeding 1000 kg disappearing, leaving only a few survivors in modern terrestrial ecosystems[24]. As a result, ungulate communities and the ecosystems they inhabit have undergone significant transformations[24,57–59]. Nevertheless, only 18% of the FTs are lost during this last episode of megaherbivore decline (Supplementary Figs. 21, 22). Although megaherbivores have been a common feature of Cenozoic faunas, they have represented only a small fraction of functional types over the last 10 Myrs, comprising 15.45% overall (15.68% in Africa, module 7; 18.18% in Eurasia, module 8; 12.5% in the Americas, module 9). Their contribution in terms of species richness remains below 14% on average (17.49% in Africa, 18.28% in Eurasia, and 6.41% in the Americas). Thus, despite megaherbivore losses, the essential functional configuration of ungulate assemblages has remained stable in terms of UFFs over the last 4.5 Myr (modules 7–9, Fig. 1).

## Summary

Our study portrays a dynamic functional system within large herbivore assemblages throughout the Cenozoic, characterized by periods of ecological stability and reassembly. Over broad temporal and spatial scales, large herbivores' functional dynamics appear mainly driven by major abiotic changes. Such events pushed the system beyond points of no-return (tipping points), forcing the reassembly into new stable states. We found two major moments of functional reassembly of terrestrial ungulate faunas that occurred synchronously across all continents. During these events, the large herbivore functional structure followed analogous trajectories in the functional space at the continental level, displaying recurrent functional homogeneity following reassembly events.

Our results show that functional systems could be resilient enough to overcome the loss of ecological roles, maintaining their functional essence during moments of environmental change, revealing the intricate relationships between functional structure and diversity. Over the last 4.5 million years, large herbivore assemblages experienced a long period of functional structure stability, overcoming even several abiotic crises, which has lasted until the present. However, this period was marked by an unchecked decline in the

functional diversity of these groups, resulting in an impoverishment of large herbivore ecosystems, notably exemplified by the loss of megaherbivores in nearly all terrestrial ecosystems across the Earth. These species performed functions in terrestrial ecosystems that are largely missing due to their extinction. Therefore, conservation efforts to preserve or reintroduce these species are essential for restoring the functionality of our ecosystems. If this functional impoverishment persists and worsens due to ongoing human impacts on the environment, it is uncertain when it will irreversibly affect the fundamental pillars of the large herbivore functional pool. This situation could potentially lead to a major third global tipping point and a new functional reassembly of ungulate faunas.

## Methods

### Database

We built a database for all the species in the largest orders of large herbivores living in earth current ecosystems (Artiodactyla, Perissodactyla, and Proboscidea) during the last 60 Ma. First, we compiled information for these species through a review of the primary literature along with data available in the NOW database[60] and the Paleobiology Database (PBDB)[60,61]. Second, following Cantalapiedra et al.[38], each species occurrence was checked in detail by experts in their taxonomic groups, looking for valid taxonomic assignments and excluding synonyms. Only records identified at the species level were retained. Third, we used the International Chronostratigraphic Chart[62] to build a consistent scheme for temporal ranges across our dataset. We checked each record in order to improve their temporal resolution through a deep survey of the literature. In some cases, there were occurrences with a broad temporal range assigned (i.e., old publications of sites with scarce stratigraphic information, early sites with a wide range of temporal assignment, or incorrect entries in the databases). When we could not find better age ranges in the literature we excluded these records from our database. We decided to exclude the occurrences with a temporal range exceeding 5 Myr in the Neogene and 10 Myr in the Paleogene. For the Plio-Pleistocene records, we manually selected the ones with broad ranges and improved them using the most recent literature.

Since we used a combination of NOW and PBDB data, we obtained duplicated records for the species in our dataset. We chose between these records based on manual, case-by-case evaluation criteria of occurrence number and overall occurrence temporal precision. As a result, after all of our criteria and review procedure was applied, we obtained a final dataset that contains 22,028 occurrences of 3046 species distributed in 10,680 sites worldwide.

Our aim was to characterize the succession of functional assemblages over evolutionary time scales. To do so, we gathered information for 13 dental traits and body size for the species in our dataset. Based on ref. 63, we compiled dental trait information for: tooth shape, degree of hypsodonty, cusp shape, number of buccal cusps, number of lingual cusps, number of longitudinal lophs, number of transverse lophs, horizodonty, as well as the presence of acute lophs, obtuse lophs, structural fortification cusps, occlusal topography, and coronal cementum (see Zliobaite et al.[46], Žliobaitė et al.[63], Fortelius et al.[64] for a detailed description of these traits). These dental traits are fundamental because they capture the relationship of the species with environmental conditions, especially being a good predictor of net primary production or precipitation[63]. Besides, we classified species in eight body size categories modified from[65,66]: B (<1 kg), C (1–10 kg), D (10–45 kg), E (45–90 kg), F (90–180 kg), G (180–360 kg), H (360–1000 kg), I (1000–10,000 kg) and J (>10,000 kg). We excluded the original category A (0–0.1 kg), because we did not have species in this body size range in our database. Overall, these traits reflect several facets of the functional role of species in ecosystems, such as habitat use, trophic level, range size, energetic requirements, and resource use[10,67].

## Spatio-temporal standardization

Working with large databases in paleontology has some difficulties, especially regarding the duplication or wrong assignment of locality' names and the heterogeneity of stratigraphic ages. In order to prevent this, when we built our occurrence matrix we merged the localities using two criteria. First, we aggregated the localities by continent (Africa, America, Asia, and Europe). To characterize macroevolutionary dynamics of Cenozoic faunas, a continental scale allowed us to track changes worldwide and check the correlation of these changes with crucial abiotic shifts in Earth' history such as plate tectonics or climate change. Second, we aggregated localities into temporal stages defined by the International Chronostratigraphic Chart stages (66, 61.6, 59.2, 56, 47.8, 41.2, 37.71, 33.9, 27.82, 23.03, 20.44, 15.97, 13.82, 11.63, 7.246, 5.333, 3.6, 2.58, 1.8, 0.774, 0.129 and 0.0117 Myr). The aggregation of localities by continent and stage yielded a total of 78 continent-stage assemblages, which are the analytical basis for the study. The use of continent-stage pools helped to avoid potential errors in the age assignments of the fossil localities.

## Characterization of ungulate function faunas

Since we consider body size and tooth shape fundamental traits to define the species' role in the ecosystems, we excluded the species with no information about these traits (this was the case for only 34 species, representing less than 0.01% of the entire species pool). The final dataset contained functional information for 3012 species, which formed the basis for all the analysis. To study the role of these species in their ecosystems, we had to reduce the dimensionality of our large mammal trait dataset. We computed Gower distances between species trait dataset using the daysy function in $R$[68]. During the computation of functional distances between species, each trait was given a different importance, so that dental traits would not have a disproportional weight in the total difference among species. To do so, we weighted two of the traits to 1 (body size and tooth shape), because they were the most representative traits of the species' relationship with their environment, and assigned a weight of 1/6 to the remaining 12 traits (hypsodonty, cusp shape, buccal cusps, lingual cusps, longitudinal lophs, transverse lophs, horizodonty, acute lophs, obtuse lophs, structural fortification cusps, occlusal topography, and coronal cementum), thus summing up to 2, which is the result of the other traits combined.

We used the resulting distance matrix in a Principal Coordinates Analysis (PCoA). Based on the functional morphospace obtained from the PCoA, we applied a *k-means* to define groups of species with similar ecological roles: functional types (FT). Species closer in the functional morphospace will be grouped in the same FT. We decided on a maximum of 406 FTs, which was the number of unique combinations of the traits in our dataset (also called functional entities Mouillot et al.[69]).

Our approach resulted in a final dataset with 406 FT distributed along 78 continent-stage pools, which were the basis for our analysis.

## Network analysis

Following Blanco, et al.[5], we used the developed Network Analysis approach to track the dynamics in the ecological assembly of large mammal continental functional pools worldwide over the last 65 Myr. We used this approach to calculate two key characteristics: affinity and fidelity. Affinity refers to the extent to which a functional type occurs in a high proportion of assemblage within a given module. In this context, a functional type with high affinity will be strongly associated with a particular module, here referred to as the ungulate functional fauna (UFF). Fidelity, on the other hand, measures the extent to which a functional type is predominantly restricted to a single module, indicating its specificity to that UFF (see methods).

**Network input data.** To analyze the temporal structuring of both functional and taxonomic communities, we implemented a community detection framework from network theory[70,71]. This technique enables the identification of clusters of fossil assemblages that exhibit similar ecological roles (functional types) or taxonomic makeup, independent of their chronological order, thus offering a window into how community structure evolved functionally over time. We translated the fossil occurrence data into bipartite networks composed of two distinct node sets: one representing continent-stage bins and the other either functional types or taxa. Connections between these nodes indicate the presence of a particular functional type (or taxon) in a given continental context. For functional networks, we further weighted these connections by counting the number of species within each functional type occurring in that bin, providing a measure of their relative prominence. In contrast, taxonomic networks were constructed using unweighted links due to the absence of data on species-level abundance.

**Community detection algorithm.** After constructing the bipartite networks, we applied the Infomap algorithm to identify communities or modules within them[72,73]. These modules correspond to sets of continent-stage bins that consistently co-occur with specific functional types or taxa, effectively grouping localities based on shared ecological or taxonomical composition. Infomap performs a joint classification of both node types−continent-stage and functional or taxonomic elements−into coherent clusters[74]. This method is grounded in the information-theoretic concept of minimum description length, which posits that the most structured representation of data is also the one that enables the highest degree of compression[75]. In our context, the regularities are expressed as modules composed of densely connected continent-stage and functional (or taxonomic) nodes, and the goal of Infomap is to identify the configuration that yields the shortest possible description of the network[73,74]. The algorithm proceeds heuristically, iteratively testing node reassignments and accepting changes that reduce the overall code length. Infomap was chosen due to its superior performance relative to other available community detection approaches[74,76]. We executed 10,000 runs of the algorithm and selected the partition with the lowest code length as the optimal solution[73]. This extensive run set ensured a comprehensive exploration of possible solutions, and the resulting modules were found to be robust across repetitions and consistent with those derived from alternative detection methods (see Sensitivity analysis to community detection).

To visualize the structure and timing of these communities, we plotted the modules identified in both the taxonomic and functional networks over geological time (Fig. 1 and Supplementary Fig. 1).

**Comparing functional and taxonomic assembly.** To examine how community assembly unfolded from both functional and taxonomic perspectives, we constructed a taxonomic counterpart to the functional dataset. This involved using the same continent-stage bins as those applied in the functional analysis, ensuring a direct comparison between both levels of organization. The resulting taxonomic dataset comprised 3046 species distributed across 78 continent-stage intervals (Supplementary Fig. 1).

**Node characterization.** We evaluated the importance of each node−whether functional type, taxon, or continent-stage−in shaping the structure of its assigned module using the IndVal index[74], which integrates two key metrics: affinity and fidelity (see Supplementary Data 1). Affinity ($A_i$) reflects how concentrated a node's connections are within its own module. It is calculated as the ratio of links a node has with others in the same module ($X_i$) to the total number of nodes in that module ($Z$):

$$A_i = X_i/Z_1 \tag{1}$$

For example, a functional type or taxon that appears frequently across continent-stages within its module will exhibit a high affinity.

Fidelity (Fi), by contrast, quantifies how exclusive those connections are to the module in question. It is defined as the number of links a node has within its module (Ni) divided by its total number of links across all modules (L):

$$Fi = Ni/L \qquad (2)$$

A node confined to continent-stages within its own module would thus have high fidelity. The final IndVal score is obtained by multiplying these two components: IndVal = Ai × Fi. High values of this index indicate nodes that are both widespread within and specific to their module, highlighting their significance in defining that community structure.

## Module characterization

We calculated the IndVal index to study the importance of the nodes in the definition of their modules. Here, we used this index to extract the most singular FTs in the modules. Through a 0.95 quantile, we extracted these singular FTs for each module. Then, we look inside these FTs to study the most abundant trait states in their species composition functional traits pool (Supplementary Figs. 4–17).

## Functional space

**Beta diversity**. In order to track the functional evolution of large herbivores, we calculated the difference in functional diversity between the different continent-stage. To do so, we used the betapart package in $R$[77]. This package calculates multi-sites distances using pairwise dissimilarities (in our case using as a basis the distance matrix between species in each bin). The advantage of using this package is that it allows to calculate the turnover and nestedness, components of functional beta diversity. Here, we used turnover as the difference of functional types composition between continent-stages, and nestedness when functional types assemblages in functional type-poor continent-stage are a subset of the assemblages in the more functional type-rich continent-stage. We ran the functional.beta.pair function, which computes functional dissimilarities based on volume of convex hull intersections in a multidimensional functional space. Dissimilarity matrix accounting for functional turnover, was measured as Simpson-derived pairwise functional dissimilarity, nestedness-resultant functional dissimilarity and functional beta diversity, was measured as the nestedness-fraction and beta diversity of Sorensen-derived pair-wise functional dissimilarity. We plotted the change in these three measures over time, calculating the change from the previous temporal bin for each continent (Supplementary Fig. 2).

We applied a non-metric multidimensional scaling (NMDS) to reduce the dissimilarity variability in two axes to plot the functional trajectories of continental assemblages over time in two dimensions. In this case, we decided to use the turnover distance matrix for the main part of the discussion and figures, as our interest is to track the functional change over time, which is the key information in order to study the dynamics of these faunas. We plotted the results using a temporal color-scale and module scale colors, obtained from the network analysis, for each continent (Fig. 1B and Supplementary Fig S19). This figure shows the different trajectories followed by the continents over time and how they explore similar regions of the functional space (Fig. 1A). The contemporaneous exploration of similar regions of the functional space (Fig. 1B) by different continents corresponds to periods of functional structure homogeneity (Fig. 1A).

## Functional diversity

Once we characterized the functional structure and its evolution using network analysis (see above), we looked into the changes in functional diversity. This allows us to study the patterns that affect the functional diversity of these faunas and compare them with the dynamics of functional structure assembly. To do so, we calculated the functional diversity for all the temporal bins separated by each continent (continent-stage) in our database. We computed this for each continent-stage based only on the functional types (i.e. groups of species with similar ecological roles, see above) belonging to the same module, from network analysis, as the continental bin.

We first use Shannon's diversity index, which considers both the richness of functional types and the even distribution of species into functional types:

$$H = - \sum R_{i=1} pi \, ln(pi) \qquad (1)$$

where R, is the number of functional types (also referred to here as functional richness), and $pi$ is the proportion of species belonging to the $i$ functional type. High values of the Shannon index indicate high continental functional richness (number of functional types) and a more homogeneous distribution of species along the functional types found in a particular continent-stage. Thus, high Shannon values indicate high values of ecological disparity (high number of functional types) coupled with high values of ecological redundancy (species evenly distributed along the functional types).

Then, we also considered each constituent part of the Shannon index (i.e. richness and evenness) independently. To measure evenness, we used the Pielou's index, calculated as,

$$J = H/\ln(R) \qquad (2)$$

where H́ comes from Shannon's diversity index. High values of evenness indicate high ecological redundancy, which means that species tend to be evenly spread across functional types (i.e. ecological roles).

For the calculation of these indexes we used only continent-stage assemblages with more than 20 species, following the sensitivity analysis to sampling performed. To account for the sampling effect of the differential number of occurrences in the different continent-stage, we extracted the residuals from linear models where indices were a function of sampling. We used the residuals from the best model (occurrences*continent) for the subsequent analysis (Supplementary Table 2).

## Linear models of functional diversity

In order to correct for the sampling effect of the differential number of occurrences in the different continent-stage[78,79], we extracted the residuals from linear models where indices were a function of sampling[80]. First, we fitted linear models where functional diversity is a function of the number of occurrences while allowing the relationship to change among continents (index ~ occurrences*continent). Second, we fitted linear models where functional diversity is a function of the number of occurrences and the quadratic of this number (index ~ occurrences*occurrences²). The best model for each index was picked based on AICc scores (Akaike's information criterion corrected for small sample size): a more complex model was only preferred if the AIC score was smaller by at least two units. For the three indexes (Shannon, Evenness and Richness), the occurrences*continent model resulted in the best model, and we used the residuals from it for the subsequent analysis (Supplementary Table 2).

Moments of abrupt physical and environmental change have been linked to reorganizations of functional structure[5]. To explore this, we fitted linear models using the corrected functional diversity indexes (see above) as a function of the continent-stage age as an explanatory variable (corrected index ~ age). We used the function *segmented* in $R$ to look for the breakpoints in the residuals of the linear model. We plotted along with the corrected index evolution through time (Fig. 2). Besides, to see how the functional diversity trends affected at geographical level and functional structure, we plotted the residuals indexes by continents through time (Fig. 2B).

### Functional distance among continents over time

In order to identify functional resemblance among continents over time, we calculated the functional distances as the distance of the continent-stage in the functional dissimilarity space. We plotted the resulting measure over time using continent pairs (Supplementary Fig. 3).

### Sensitivity analysis

**Sensitivity analysis on the k-means groups.** We tested the selection criterion of 406 FT based on the number of unique combinations of functional traits. We ran the k-means analysis selecting 100, 200, 300, 400 and 500 groups, and we repeated 100 times. We used this to build the analysis database and run infomap[72]. As a result, we plotted the quality code length (measure of modularity) and the number of modules obtained from the network analysis for all k-means randomization (Supplementary Fig. 18). In both cases the variable values become stable around 400 FT.

**Sensitivity analysis to sampling.** When working with paleontological data, we should always be aware of potential sampling biases that could drastically affect the results of our analyses (i.e. some epochs being better sampled, countries or regions with difficult access to conduct fieldwork, historical preferences…). In other words, the disparate number of species across the analyzed bins could impact the estimated functional similarity between them. To test this, we conducted a sensitivity analysis where we performed a rarefaction procedure[79,80] on the species present in each continent-stage. We randomly selected 20 species from each bin, and then we ran the network and beta diversity analysis (see above for detailed explanation) on the rarefied dataset. We repeated the procedure 100 times. An averaged distance matrix from the 100 draws was used to estimate functional turnover between bins and to compare with our main results (Supplementary Fig. 19). The results from the rarefactions were highly congruent with those derived from the full dataset.

**Sensitivity analysis to the effect of aggregation.** To determine whether the modular patterns found in our functional networks were genuinely reflective of ecological structure—or merely an artifact of grouping species into functional types—we performed a null model analysis. Specifically, we generated randomized networks in which species were reassigned to functional types at random, while preserving the original number of species per functional type. We then compared the modularity of these null networks to that of the observed network. Modularity was assessed within the Infomap framework by calculating the relative code length: the ratio of the code length for the partitioned network (CLm) to the code length of the same network without any modular structure (CL). A relative value close to 1 indicates low modularity, whereas values near 0 suggest strong community structure. To express modularity in a more intuitive way, we used its complement:

$$M = 1 - CLm/CL \qquad (3)$$

Here, CLm refers to the code length of the modular network, and CL to that of the unpartitioned network. To assess statistical significance, we computed a *p*-value based on the proportion of 100 randomized networks (plus the observed one) that showed modularity equal to or greater than that of the empirical data (Supplementary Fig. 20).

**Sensitivity analysis to community detection.** We evaluated two main sources of uncertainty in our detection of network communities. First, since algorithms like Infomap rely on stochastic optimization to identify network partitions, multiple runs can yield slightly different solutions. This raises two important questions: how many runs are needed to reliably capture the full landscape of possible solutions, and how stable the resulting modules are across near-optimal partitions[81]. To assess the first, we conducted a 10-fold cross-validation, estimating the likelihood that a randomly held-out partition would share at least 75% similarity with one from the training set. Results showed that executing Infomap 10,000 times yielded a nearly complete exploration of the solution space, with test partitions almost always exhibiting similarity above 0.75—both for functional and taxonomic networks.

To evaluate the second aspect—module stability—we examined whether the communities found in the best partition reappeared in the rest of the 9999 solutions. For each module, we computed the probability of retrieving a similarly composed group in other runs, using two thresholds of similarity: a standard one (0.5), implying one-to-one correspondence, and a stricter threshold of 0.75[81]. In both cases, the core modules remained highly reproducible across solutions, with probabilities close to 1, indicating strong consistency (Fig. 1 and Supplementary Table 1).

**Sensitivity analysis to accuracy of network partition vs functional distances.** We ran a Permutational multivariate analysis of variance (PERMANOVA) to test whether the module partition from network analysis reflects the functional distances between the continent-stage. We obtained that the module partition is similar to functional distances by a statistically significant $p > 0.001$.

### Potential limitation of the spatio-temporal standardization

Our approach focuses on aggregating localities into a spatio-temporal sequence that allows us to study the evolution of the functional structure of large mammals at the continental level. We acknowledge the limitations of this approach, particularly regarding spatial standardization. A finer-scale geographic partitioning (e.g., by country, region, or grid) could potentially yield different functional structure patterns. However, these results would reflect a different spatial scale and should be interpreted accordingly. Our findings capture broad continental-level patterns, which remain consistent across all sensitivity analyses performed (see above).

One limitation of our study is the merging of North and South America in the functional analyses. Due to the limited availability of data (both in occurrences and functional traits) for South American species (only 50 taxa), analyzing the continents separately would have resulted in highly uneven datasets, potentially biasing comparisons. South America's fossil record is less extensively studied than North America's, mainly due to preservation biases in tropical regions, further constrained by geopolitical and economic factors. Additionally, certain endemic South American herbivore groups, such as notoungulates, have been less studied due to the scarcity and fragmentary nature of their fossil record.

While this approach allows for more robust statistical comparisons, we acknowledge that North and South America had largely independent faunal histories prior to the Great American Biotic Interchange (GABI). The merging of these datasets may obscure some biogeographic patterns, particularly those associated with faunal exchange events. Future work incorporating a more complete South American fossil record, including a broader representation of endemic taxa, will help refine our understanding of the dynamics of functional diversity across both continents.

We used *R* software[68] for all the analyses. All the scripts and database used are at: https://github.com/f-blanco/ecoherb and refs. 82,83.

### Reporting summary

Further information on research design is available in the Nature Portfolio Reporting Summary linked to this article.

## Data availability

Source data for all figures are provided in the accompanying Source Data file. All data used in this study is available at:https://github.com/f-blanco/ecoherb and https://figshare.com/authors/Fernando_Blanco/9731162[82]. Primary data were originally sourced from: https://paleobiodb.org and https://nowdatabase.org Source data are provided with this paper.

## Code availability

All code used for the analyses is available at: https://github.com/f-blanco/ecoherb and https://figshare.com/authors/Fernando_Blanco/9731162[83].

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

## Acknowledgements

We thank M. Hernández Fernández, P. Medina-García, I. Menéndez, G. Navalón, the Amniota lab and Computational Paleobiology lab for their valuable comments and advice. We also thank A. B. Shupinski and two anonymous reviewers for their comments, which helped us improve this manuscript. F.B. was funded by Gothenburg University via the Swedish Research Council (VR: 2019–04739) and F.B., J.L.C. and J.M. was funded by the Deutsche Forschungsgemeinschaft (LO 2368/1–1). J.L.C. was funded by the Talent Attraction Program of the Madrid Government (no. 2017–T1/AMB5298) and the grant CNS2023–144573 funded by MCIN/AEI/10.13039/501100011033 and by the European Union NextGenerationEU/PRTR. The author I.A.L acknowledges funding from a Postdoctoral Research Fellowship from the Humboldt Foundation and a Ramon y Cajal Fellowship from the Ministerio de Ciencia e Innovación de España under grant No. RYC2021–034991-I, funded by MCIN/AEI/10.13039/501100011033 and European Union NextGenerationEU/PRTR.

J.S. was funded by the Research Council of Finland, project nr. 340775. D.S. received funding from ETH Zurich, the Swedish Research Council (VR: 2024-04303), and the Swedish Foundation for Strategic Environmental Research MISTRA within the framework of the research programme BIOPATH (F 2022/1448). M.R. thanks the Stimulus of Scientific Employment, Individual Support – 2018 Call grant by the Fundação para a Ciência e a Tecnologia (Portugal, CEECIND/02199/2018) and GeoBioTec (UIDB/04035/2020, DOI: 10.54499/UIDB/04035/2020). We acknowledge the effort of all the people working in fossil sites over the decades that made this study possible.

## Author contributions

F. Blanco, J.L.C. and J.M. conceptualized the research. F. Blanco, I.A.L, O.S, F. Bibi, N.S.H, M.R, B.M., M.T.A., J.L.P, J.S. and J.L.C gathered the occurrence and trait data. F. Blanco, J.C., D.S and J.L.C. designed the analysis. F. Blanco performed the analysis. F. Blanco, and J.L.C. wrote the paper with input from all authors.

## Funding

## Competing interests

The authors declare no competing interests.

## Additional information

[1]Department of Biological and Environmental Sciences, University of Gothenburg, Medicinaregatan 7B, 413 90 Gothenburg, Sweden. [2]Gothenburg Global Biodiversity Centre, 405 30 Gothenburg, Sweden. [3]Museum für Naturkunde, Leibniz Institute for Evolution and Biodiversity Science, 10115 Berlin, Germany. [4]National Research Center on Human Evolution (CENIEH), Paseo Sierra de Atapuerca 3, 09002 Burgos, Spain. [5]Department of Evolution, Ecology, and Behaviour, University of Liverpool, Liverpool, United Kingdom. [6]Global Change Ecology and Evolution Research Group, Department of Life Sciences, Universidad de Alcalá, GloCEE, Madrid, Spain. [7]GeoBioCenter LMU, Ludwig-Maximilians-Universität München, München, Germany. [8]Department of Earth and Environmental Sciences, Palaeontology and Geobiology, Ludwig-Maximilians-Universität München, München, Germany. [9]GEOBIOTEC, Department of Earth Sciences, NOVA School of Science and Technology, Universidade NOVA de Lisboa, Campus de Caparica, Caparica, Portugal. [10]Museu da Lourinhã, R. João Luís Moura, Lourinhã, Portugal. [11]Naturhistorisches Museum Basel, Basel, Switzerland. [12]Departamento de Paleobiología, Museo Nacional de Ciencias Naturales (CSIC), Madrid, Spain. [13]Universidad Nacional del Centro de la Provincia de Buenos Aires. Del Valle, Olavarría, Argentina. [14]Department of Geosciences and Geography, University of Helsinki, Helsinki, Finland. [15]Department of Biosystems Science and Engineering, ETH, Zurich, Switzerland. [16]Universidad Rey Juan Carlos, Calle Tulipán s/n, Madrid, Spain. [17]Global Change Research Institute, Rey Juan Carlos University, Madrid, Spain. ✉e-mail: fblancosegovia@gmail.com

