## [Transparent Peer Review file · Nature Communications]

Two major ecological shifts shaped 60 million years of ungulate faunal evolution

Corresponding Author: Dr Fernando Blanco

Version 0:

Reviewer comments:

Reviewer #1

(Remarks to the Author)

This is a very interesting paper, where the authors use an approach based on network analyses for studying the faunal dynamics of ungulate continental assemblages (orders Artiodactyla, Perissodactyla and Proboscidea) during the Cenozoic and their reorganization in response to major tectonic and climatic events.

They find (i) a prevalence of long periods of ecological stasis; (ii) a decoupling of taxonomic and functional assembly dynamics; and (iii) two major shifts (tipping points) in the functional diversity of the guilds of primary consumers at 26-21 Ma (i.e., at the Oligocene-Eocene transition) and after 10 Ma (i.e., at late Miocene times), respectively.

According to their interpretation, the first tipping point relates to the formation of a land bridge between Eurasia and Africa, which led to a transition towards a prevalence of large browsers with mid- to high-crowned molars. The second resulted from aridification and the spread of C4-dominated vegetation, which triggered the emergence of an ungulate fauna characterized by grazers and browsers with high and low crowned teeth, respectively.

The authors cite a couple of papers by Figueirido et al. in PNAS (their refs. 10 and 11) as examples of studies addressing faunal shifts in the fossil record that "have been traditionally tackled from a taxonomic perspective". However, this applies to Figueirido et al. (2012), who analyzed the evolutionary faunas of the North American Cenozoic using a taxonomic perspective, but not to Figueirido et al. (2019), a paper where the approach was based on a taxon-free ecomorphological characterization of these evolutionary faunas. Moreover, the latter study included analyses of functional diversity of large mammals (not only ungulates) based on tooth types, as those performed in the current study by Blanco and colleagues, but also on the frequencies of locomotor types and body size classes among mammalian assemblages, an approach that provided a wider and more balanced view of the changes through time in ecomorph diversity. While acknowledging that ungulates are primary "ecosystem engineers", it would be worth considering here if a better picture of past ecosystems could emerge from an analysis of the whole community of large mammals.

Finally, a couple of relatively minor points that should also be addressed:

1.- After the Mid-Miocene Climatic Optimum, the transition towards a drier and colder climate drove the development of hypsodont teeth in ungulates (and also of hypercursorial adaptations, as noted by Figueirido et al., 2019), which allowed them to process the more abrasive foodstuffs available in open habitats. However, this dietary adaptation is related to (i) the presence of external abrasives (grit) on the plants from open habitats and (ii) the consumption of soil rather than to the presence of internal abrasives (silicophytoliths) in the herbaceous plants, as demonstrated by Mendoza & Palmqvist (2008; *J. Zool.*, 274: 134-142) and Damuth & Janis (2011; *Biol. Rev.*, 86: 733-758).

2.- The authors comment on only one major dispersal event between Eurasia and Africa at the Oligocene-Eocene transition. However, in their analysis of North American evolutionary faunas of large mammals Figueirido et al. (2012) also indicated that the rise of the Oligocene fauna may relate to the major immigration event from Asia in North America, which took place during the middle-late Eocene (i.e., at 37.5 Ma) following a sea level fall at this time. Also, they indicated that the rise of the North American Pliocene fauna may be related with a major immigration that occurred near the end of the early Miocene, which suggests that this evolutionary fauna began to diversify at approximately 16.5 Ma, shortly after an extensive interchange between Eurasian and North American faunas, and that this immigration event may also have played a role in the decline of the Miocene fauna.

Reviewer #2

(Remarks to the Author)

I found the manuscript was well-written, concise and enjoyable to read. I believe it would be a great article for Nature Communications to publish. I felt it conveyed important findings about macroevolutionary changes in ungulates. Results and discussion were kept broad and logical without over-inferring driving impacts. The story was upheld of investigating continental scale changes through time without getting lost in the details. The figures are limited to those providing key information and are visually appealing. The methods in functional diversity and network analysis are sound and possible biases in the data were addressed appropriately in the supplementary material.

There are a few points I would like addressed before acceptance. I added to my review document including the combination of North and South America, missing traits, the clear definition of reconfiguration vs. reassembly and the interpretation of high functional diversity. I feel confident these concerns can be appropriately addressed with logical explanations and/or additional analyses.

Thank you for your contribution to science.

Reviewer #3

(Remarks to the Author)

Dear authors,

I think that this article is very important and meet some vital aspects required for a good article. The methods are clearly stated and adapted to the question. Their limitations are considered and the author's team overcame them with classic corrections methods. The dataset build is not new, it's made of existing data from various sources (databases, literature) but they use it at a very large scale to interrogate crucial questions. How functional diversity has changed during the last 60 million years for ungulate all around the globe? By using various classical methods (e.g. clustering, models, ordination) on network of functional type they were able to detect significant breaks and tendencies in the continuum at global and continental scales. They discovered timing of emergence of modern patterns and tried to link these breaks, emergence and tendencies with known abiotic and biotic events. I really enjoyed this paper and congratulate the authors for this quality research. After answering to reviewer remarks and questions, I would recommend a minor revision. My remarks are dedicated to make the paper easier to understand, I would try to highlight the parts where I struggled and suggest potential arguments to make a better case of the observed patterns.

Major remarks

Figures are very well done and most of them are easy to interpret and understand. However, I struggled to really understand Fig. S1. The caption could be extended to better explain its construction and/or interpretation. I had difficulties to understand the statement made about it line 66 ("Our results show a decoupling of taxonomic and functional") while looking at the figure. Please, either extend the caption, or add some text to interpret this figure in the page 1 of the supplementary data (because I'm sure you have to keep the main text as concise as possible). I think that the statement made line 68-69 is very important (even if already known in others context/studies) and could be more easily understood if the description of Fig. S1 would be extended in supplementary files.

L. 291-293: "The aggregation of localities by continent and stage yielded a total of 78 continent-stage assemblages, which are the analytical basis for the study". I think that's totally needed and fair for a study at the temporal and spatial scale, but could you develop a bit the potential limitation of such approach in your method section (or in supplementary)? What could be the impact of such data agglomeration? For example, could different UFFs co-exist on one continent (e.g. SE Asia vs Central Asia) and thus create fake stability by experiencing different trajectories of changes through times?

Minor remarks

I think that the authors could have spend more time to compare known patterns of taxonomic diversity in mammals that took interest in clustering temporal fauna (e.g. chronofauna in both North America and Europe/Asia/Africa, see for example Figueirido, Gibert, Casanovas-Vilar) and look for breaks and tendencies but I understand that Nature format is very constrained. However, I think that the excellent work they did with functional diversity could have been even better if it would have been better compared to other dimensions of mammalian fossil biodiversity.

L. 74-75 : "Since 10 Ma, large herbivore assemblages have experienced remarkable stability".

One remark: it could be stated here that Asian UFFs are acting slightly differently than the others continental fauna.

One question: when looking at Fig. 1B, we can see how the average/median point of UFFs in NMDS is moving in the environmental space through time, but I struggle to understand how Fig. 1A and Fig. 1B relate.

For example, when looking at the orange/yellow section for Africa (i.e. the last 10 myr), we see strong displacement in the environmental space for the african UFF, but it corresponds to stasis (or a single mode, turquoise-green) in Fig. 1A. In the same way, the early Cenozoic asian UFF relate to the first mode (i.e. red) and last for ~35 myr, but we can see in Fig. 1B this UFF moving a lot through the environmental fauna (i.e. black to red section of the line). Can you extend the explanation in supplementary data to understand the relationship between Fig. 1A and 1B. Fig. S19 could be used to explain this relationship with more details.

L. 97-98 : "This was primarily driven by the emergence of functional dominance of FTs in larger body size categories in the Americas, in contrast to the mid-sized herbivores that characterize the UFF of Eurasia (Fig. S17)".

Could it be a matter of the relationship between landmass and (paleo)latitudes? The larger southern extent of Asia could explain the survival of mid-sized herbivores and the continuous cooling of Cenozoic could act in very different direction depending on the latitudinal distribution of landmass, isn't it? The argument developed in the rest of this section (from L.95 to L.119) could be completed by a discussion around landmass/biome (i.e. closed vs open forests) distribution. The two references associated with EOC transition (29, 40) are associated with very descriptive arguments (L.109-119) when explanatory argument (i.e. mechanism) could/should be invoked here (e.g. Latitude and landmass, Europe being restricted to high latitude and forming an archipelago making climate tracking more difficult for European fauna). On a more general statement, I think that this paper is sometimes a bit too descriptive, more potential mechanisms could be invoked to explain the observed patterns.

L. 124-125: "The functional coalescence is reflected in the drastic reduction of the functional distance between Africa and Eurasia after 21 Ma".

Very interesting, could you just add the age uncertainties (i.e. the corresponding geological stage length if I correctly understood your methodology) in order to make this statement more subtle. The exact date of the main part of the interchange/faunal mixing between Africa and Eurasia remain debated and your sentence could imply that your study can precisely date this event.

L. 298: "We excluded the species with no information about these traits".

Did you analyse/reanalyse/measure species traits or did you find the 13 traits being already described for the 3012 species included in your study?

L. 306: "and assigned a weight of 1/6 to the remaining 12 traits".

Have you tried different weight, in other word, did you perform sensitivity analysis on this parameter ?

L. 393-397: Thank for the clear interpretation of the Shannon index for FD. Would you say that high Shannon values in your study relate to strong ecosystem stability (high diversity + high redundancy)? The comparison of Shannon FD and Fig. 1 is not trivial.

Thank you again for your future answers and congratulation again for this very interesting study.

Version 1:

Reviewer comments:

Reviewer #2

(Remarks to the Author)

The authors find major shifts in the functional diversity of ungulates over the 60 million years associated with ecological and environmental shifts. They also find a decoupling between taxonomic and functional diversity fluctuations through time. The research and methodology are sound and appropriate for the questions being asked by the authors. I continue to believe that this manuscript provides important findings about the dynamics of mammals over evolutionary timescales, providing a baseline for pre-human ecology. Moreover, I believe it will be of interest to a wide audience.

After reviewing the minor revisions and responses to the reviewers, I have no further concerns or comments regarding the manuscript and would like to see this published in Nature Communications.

(Remarks on code availability)

The trait and occurrence data is available for analysis and review. All necessary details are included and easy to interpret. The code is clean and easy to follow regarding reproducibility. Therefore, I feel that the provided code and data is acceptable for publication.

Reviewer #3

(Remarks to the Author)

Dear authors,

Thank you for taking into account all the suggestions, and modifying your text accordingly.

(Remarks on code availability)

Response to reviewers comments

Reviewer comments are highlighted in **bold**.

Responses to reviewers are provided in regular text, following each respective comment.

Reviewer #1:

This is a very interesting paper, where the authors use an approach based on network analyses for studying the faunal dynamics of ungulate continental assemblages (orders Artiodactyla, Perissodactyla and Proboscidea) during the Cenozoic and their reorganization in response to major tectonic and climatic events. They find (i) a prevalence of long periods of ecological stasis; (ii) a decoupling of taxonomic and functional assembly dynamics; and (iii) two major shifts (tipping points) in the functional diversity of the guilds of primary consumers at 26-21 Ma (i.e., at the Oligocene-Eocene transition) and after 10 Ma (i.e., at late Miocene times), respectively. According to their interpretation, the first tipping point relates to the formation of a land bridge between Eurasia and Africa, which led to a transition towards a prevalence of large browsers with mid- to high-crowned molars. The second resulted from aridification and the spread of C4-dominated vegetation, which triggered the emergence of an ungulate fauna characterized by grazers and browsers with high and low crowned teeth, respectively.

We sincerely appreciate the positive feedback from Reviewer #1 regarding our methodological approach and key findings.

The authors cite a couple of papers by Figueirido et al. in PNAS (their refs. 10 and 11) as examples of studies addressing faunal shifts in the fossil record that “have been traditionally tackled from a taxonomic perspective”. However, this applies to Figueirido et al. (2012), who analyzed the evolutionary faunas of the North American Cenozoic using a taxonomic perspective, but not to Figueirido et al. (2019), a paper where the approach was based on a taxon-free ecomorphological characterization of these evolutionary faunas.

We appreciate the reviewer's comment and acknowledge that Figueirido et al. (2019) employed a taxon-free ecomorphological characterization of evolutionary faunas. The inclusion of this reference in that context was an oversight, and we have corrected it by removing Figueirido et al. (2019) from the citation.

Moreover, the latter study included analyses of functional diversity of large mammals (not only ungulates) based on tooth types, as those performed in the current study by Blanco and colleagues, but also on the frequencies of locomotor types and body size classes among mammalian assemblages, an approach that provided a wider and more balanced view of the changes through time in ecomorph diversity. While acknowledging that ungulates are primary “ecosystem engineers”, it would be worth considering here if a better picture of past ecosystems could emerge from an analysis of the whole community of large mammals.

We appreciate the reviewer's insightful comment regarding the potential benefits of analyzing the entire large mammal community. However, we focus on ungulate faunas for several reasons:

1. Ecological significance: Ungulates have played a dominant role in shaping terrestrial ecosystems since the K-Pg mass extinction, acting as primary ecosystem engineers (Brits et al 2002, van der Waal et al. 2011, Smith et al. 2016).
2. Fossil record availability: Their widespread distribution and abundant fossil remains throughout the Cenozoic enable a large-scale macroevolutionary analysis (Zliobaite and Fortelius 2022). In contrast, other mammal groups (e.g., Carnivora) tend to have a more limited and sparse fossil record, making a comparable analysis challenging.
3. Temporal continuity: We focus on the orders *Artiodactyla*, *Perissodactyla*, and *Proboscidea* because they are the only large herbivore clades with both extant representatives and a fossil record extending back 60 million years. This continuity allows us to analyze long-term trends in functional structure.
4. Dataset scope: Thanks to these factors, we assembled a robust dataset comprising 22,028 occurrences of 3,046 species from

10,680 fossil sites worldwide, providing an unprecedented temporal and spatial resolution for studying functional diversity in large herbivores.

While incorporating additional mammalian groups could offer a broader perspective on ecosystem dynamics, the limitations in their fossil representation would compromise the feasibility and comparability of such an analysis. Future progress in taxonomic and systematic research, along with new fossil discoveries from ongoing and future excavations, will be crucial for expanding our understanding of the entire large herbivore guild and even mammals as a whole. As more data become available, such analyses may become feasible, potentially offering deeper insights into past ecosystems.

1.- After the Mid-Miocene Climatic Optimum, the transition towards a drier and colder climate drove the development of hypsodont teeth in ungulates (and also of hypercursorial adaptations, as noted by Figueirido et al., 2019), which allowed them to process the more abrasive foodstuffs available in open habitats. However, this dietary adaptation is related to (i) the presence of external abrasives (grit) on the plants from open habitats and (ii) the consumption of soil rather than to the presence of internal abrasives (silicophytolithes) in the herbaceous plants, as demonstrated by Mendoza & Palmqvist (2008; J. Zool., 274: 134-142) and Damuth & Janis (2011; Biol. Rev., 86: 733-758).

We appreciate the reviewer's insightful comment regarding the factors influencing dental abrasion in ungulates. To address this, we have incorporated the relevant information on line 190, emphasizing the role of external abrasives (grit) and incidental soil consumption in driving the evolution of hypsodonty:

"In this new context, with the need to cope with more abrasive particles in their diet (from plant tissues, soil and grit on the plants), several herbivore lineages developed new dental functional structures and traits, including increased relative tooth height (hypsodonty), larger molar occlusal areas, flatter occlusal topography or increasing number of cutting, transversal lophs, among others (45-50)."

These factors played a crucial role in the development of increased tooth height, larger molar occlusal surfaces, flatter occlusal topography, and a greater number of transverse lophs, among other adaptations in ungulate faunas.

2.- The authors comment on only one major dispersal event between Eurasia and Africa at the Oligocene-Eocene transition. However, in their analysis of North American evolutionary faunas of large mammals Figueirido et al. (2012) also indicated that the rise of the Oligocene fauna may relate to the major immigration event from Asia in North America, which took place during the middle-late Eocene (i.e., at 37.5 Ma) following a sea level fall at this time. Also, they indicated that the rise of the North American Pliocene fauna may be related with a major immigration that occurred near the end of the early Miocene, which suggests that this evolutionary fauna began to diversify at approximately 16.5 Ma, shortly after an extensive interchange between Eurasian and North American faunas, and that this immigration event may also have played a role in the decline of the Miocene fauna.

We thank the reviewer for these comments. We have incorporated the suggested information into line 94: *“Functional innovations were likely intertwined with biogeographic events, contributing to this enrichment. For example, around 37.5 Ma, a significant immigration of large mammals from Asia to North America occurred following a sea-level drop (10).”*

And line 165: *“Interestingly, this decline follows an immigration event between Eurasia and North America at the end of the early Miocene, some 16.5 Ma (10), suggesting that faunal dispersals do not necessarily enhance functional diversity but may instead reduce the functional richness of endemic American functional types.”*

Reviewer #2:

I found the manuscript was well-written, concise and enjoyable to read. I believe it would be a great article for Nature Communications to publish. I felt it conveyed important findings about macroevolutionary changes in ungulates. Results and discussion were kept broad and logical without over-inferring driving impacts. The story was upheld of investigating continental scale changes through time without getting lost in the details. The figures are limited to those providing key information and are visually appealing. The methods in functional diversity and network analysis are sound and possible biases in the data were addressed appropriately in the supplementary material. There are a few points I would like addressed before acceptance. I added to my review document including the combination of North and South America, missing traits, the clear definition of reconfiguration vs. reassembly and the interpretation of high functional diversity. I feel confident these concerns can be appropriately addressed with logical explanations and/or additional analyses.

Thank you for your contribution to science.

We sincerely appreciate the reviewer's thoughtful and encouraging comments on our manuscript. We are pleased that they found the manuscript well-written, concise, and engaging, and that our approach to studying macroevolutionary changes in ungulates was clear and appropriately framed. We also appreciate the reviewer's positive assessment of our methodological framework and data considerations.

We provide detailed responses below, along with clarifications and changes in the manuscript.

Introduction

Lines 48 – 53: I think it would be helpful for the geographic distribution to be stated somewhere in the description of the data here. As a reader, it was the first question that popped in my head. Just adding “global” would be sufficient.

Ok, we added “global” in that sentence.

Results and Discussion

Lines 78: Can you clarify why North and South America were analyzed together? Due to the independence of the two faunas, it seems more reasonable that they would be analyzed separately. It seems important to note if their separation alters the overall functional structure. Moreover, you would potentially expect a shift at GABI.

We merged North and South America in our analysis due to limited data availability for South America, particularly for the three focal orders (Artiodactyla, Perissodactyla, and Proboscidea), which include only 50 species there. Several factors explain this limitation: 1) The fossil record of South American mammals is not as extensively studied as that of North America, which benefits from a long tradition of paleontological research. Biases in the fossil record of South America, particularly in tropical regions, as well as geopolitical and economic factors, contribute to this disparity. 2) South America hosted other large herbivore groups, such as notoungulates, which were not included in our analysis. We acknowledge that future excavations and ongoing studies on endemic South American herbivores may allow for their inclusion in future analyses, providing a more comprehensive understanding of the ecosystems in which these groups lived. We have added text acknowledging this limitation in the new section “Potential limitations of the spatio-temporal standardization”.

Lines 68 – 73: What defines a functional reconfiguration? And how does that vary from tipping points? You say there is a functional reconfiguration every ~6 Myr, but there are long periods of ecological stasis and there are two periods of reassembly. Could you clarify how those different types of events are defined? I became confused when you said “with a first reconfiguration in UFFs around 40 Ma (line 86)

We define *functional reconfiguration* as any module change identified through network analysis. *Tipping points*, on the other hand, refer to moments when these changes occur simultaneously across all

continents (i.e., a simultaneous module change worldwide). To clarify this distinction, we revised the sentence in line 71:

“Large herbivore assemblages experienced long periods of ecological stasis punctuated by two episodes of global and almost synchronous functional reassembly (tipping points, Fig. 1A). These tipping points occurred around 21 and 10 Ma, likely as a response to major abiotic events. Since 10 Ma, large herbivore assemblages have experienced remarkable stability.”

In this context, we mean that the functional structure underwent a module change every ~6 million years on average, while the taxonomic structure changed more rapidly, with a module change every 1.7 million years on average. We revised the sentence for clarity as follows in line 67:

“On average, taxonomic reassembly (module change, see Fig. S1) occurred every 1.7 ± 0.31 Myr, whereas functional reassembly took place every 6.25 ± 1.6 Myr. “

Regarding UFFs (Ungulate Functional Faunas), we have clarified their definition since the introduction: ungulate continental assemblages with shared configurations (presence, relevance, and fidelity) of functional types (see line 67):

“We then applied Network Analysis to identify ungulate continental ungulate functional faunas (UFFs)(5), defined as unique associations of functional types. Notably, UFFs are not defined solely by the presence of FTs, but by their relevance, affinity, and fidelity in the faunas (see supplementary methods).”

UFFs are the modules in our network analysis (see Fig. 1A). The first functional reconfiguration in the UFFs (module change) occurred around 40 Ma, coinciding with the split of the Americas from the Eurasian module.

We hope that these clarifications make the terminology more precise and easier to follow.

Line 149: I would replace “recovered” with identified. I had to read it twice because I thought you were suggesting that functional diversity was recovering in those continents.

Ok, we have replaced "recovered" with "identified" to avoid any confusion.

Lines 197 – 198: If you were looking at functional diversity of North and South America together before GABI, you wouldn’t suspect a change in the overall functional pool following the interchange.

As the reviewer pointed out, the functional pool (i.e., functional richness, or the number of functional types) was not the main driver of the functional diversity increase (Fig. 2B). In fact, our results indicate a decrease in functional diversity during that period. The observed increase in functional diversity is primarily due to a rise in functional evenness (Fig. 2B). This suggests that after the GABI, there was an increase in redundancy, with species becoming more evenly distributed across functional types (i.e., ecological roles). While functional richness decreased, the new species that evolved during this period occupied similar ecological roles to those that existed previously.

Lines 198 – 199: The functional structure has remained the same following all the extinctions and extirpations? If you mean in the most recent interval (encompassing the last 10,000 years), I would say that, not until present day. The way I understand it, is all large ungulates that have existed on the continent during any part of that interval is included in determine the functional structure.

Our results show a continuous decline in functional diversity from 4.5 Ma, primarily driven by a decrease in functional richness. However, functional evenness slightly increased during this period. While functional types are being lost, new species that evolved during this time occupied pre-existing functional roles, leading to increased redundancy and potentially contributing to the stability of the functional structure (modules).

For a functional reconfiguration (module change) to occur, it is not only necessary for functional types to go extinct, but these extinctions must involve types that are highly connected within a module, with minimal connections to other modules (high affinity and fidelity, showing a high IndVal index value, see method section in the supplementary). These types are critical for defining modules due to their exclusive connections within specific groups. Furthermore, a recent study using a different network analysis approach (bayesian network analysis of food webs) found persistent stability throughout the Pleistocene in Eurasian mammals (Bekeraite et al. 2024), supporting our findings at a global scale.

Methods

Lines 297 – 298: It is unclear to me if “no information about these traits” means the species needed to be missing all of the traits or if they are missing a information about a single trait. If species were kept that were missing 1 or multiple traits, could this bias your data and was a sensitivity analysis used to address any possible biases?

We discarded species lacking data on body size or tooth shape, as these traits are fundamental to defining the ecological roles that species played in their ecosystems. Only 35 species out of more than 3,000 were missing these traits, representing less than 0.01% of the entire species pool. We believe this minimal exclusion does not introduce significant bias to our analysis. Although no sensitivity analysis was conducted specifically for this issue, given the small proportion of species affected, we do not expect this omission to substantially impact the results.

For clarity, we have slightly rephrased the sentence as follows:

“Since we consider body size and tooth shape fundamental traits to define the species’ role in the ecosystems, we excluded the species with no information about these traits (this was the case for only 34 species, representing less than 0.01% of the entire species pool). “

Lines 395 – 397 and 404 - 406: a homogenous distribution in functional space does not necessarily represent high redundancy.

It is possible to get lower redundancy and an increase in distribution homogeneity across functional space. So I'm not sure I believe high Shannon values always represent ecological redundancy, unless I am misunderstanding something that addresses this in the equation. Is there any reason to think that geographic distribution of available occurrences could influence your results during any time periods?

To calculate the Shannon index, we did not use the distribution in functional space. Instead, we applied the classical Shannon index equation, which incorporates both richness and evenness. High values of the Shannon index require both a high number of functional types and a high degree of species evenly distributed among these types.

To account for potential biases in the geographic distribution of occurrences, we calculated the Shannon index for each continent-stage using only the functional types (i.e., groups of species with similar ecological roles) within the same module, as identified by network analysis, corresponding to the continental bin. Additionally, we limited our analysis to continent-stage assemblages with more than 20 species, as guided by the sensitivity analysis on sampling.

To further address sampling effects related to the differential number of occurrences across continent-stages, we extracted the residuals from linear models where the indices were modeled as a function of sampling (see Supplementary Methods). The residuals from the best-fitting model (occurrences*continent) were then used in the subsequent analysis (Table S3).

All procedures are described in detail in the Methods section of the main text and the Supplementary Methods.

Figures

In Fig. 1A African data begins around 30 Ma, in Fig. S2 it begins around 25 Ma and in Fig. 2, African data does not start until 20 Ma. Can you explain why?

Before 30 Ma, African assemblages only included proboscideans. Therefore, we decided to exclude the earlier part of the African Cenozoic

(60-30 Ma) from the analysis to maintain a comparable framework with the other continents.

Regarding Fig. S2, the beta diversity difference is calculated by the difference between one bin and the previous one. Since the first bin starts at 30 Ma, we discarded this bin and began the calculation with the following one (Chattian stage with a mean age of 25 Ma).

For Fig. 2, as explained above and detailed in the Methods and Supplementary Methods sections, we applied a filter requiring at least 20 species per continent-stage for inclusion in the analysis. The first African time bins did not meet this condition, so they were not included. The first bin that satisfied this condition is at 20 Ma (Burdigalian stage).

Reviewer #3:

Dear authors,

I think that this article is very important and meet some vital aspects required for a good article. The methods are clearly stated and adapted to the question. Their limitations are considered and the author's team overcame them with classic corrections methods. The dataset build is not new, it's made of existing data from various sources (databases, literature) but they use it at a very large scale to interrogate crucial questions. How functional diversity has changed during the last 60 million years for ungulate all around the globe? By using various classical methods (e.g. clustering, models, ordination) on network of functional type they were able to detect significant breaks and tendencies in the continuum at global and continental scales. They discovered timing of emergence of modern patterns and tried to link these breaks, emergence and tendencies with known abiotic and biotic events. I really enjoyed this paper and congratulate the authors for this quality research. After answering to reviewer remarks and questions, I would recommend a minor revision. My remarks are dedicated to make the paper easier to understand, I would try to highlight the parts where I struggled and suggest potential arguments to make a better case of the observed patterns.

We sincerely appreciate the reviewer's detailed and positive feedback on our manuscript. We are pleased that they found our study important, methodologically sound, and relevant to understanding long-term changes in ungulate functional diversity.

We have carefully addressed each of their points and incorporated their suggestions where applicable to strengthen the presentation of our findings.

Figures are very well done and most of them are easy to interpret and understand. However, I struggled to really understand Fig. S1. The caption could be extended to better explain its construction and/or interpretation. I had difficulties to understand the statement made about it line 66 ("Our results show a decoupling of taxonomic

and functional”) while looking at the figure. Please, either extend the caption, or add some text to interpret this figure in the page 1 of the supplementary data (because I’m sure you have to keep the main text as concise as possible). I think that the statement made line 68-69 is very important (even if already known in others context/studies) and could be more easily understood if the description of Fig. S1 would be extended in supplementary files.

Thank you for pointing this out. We have expanded the caption of Fig. S1 to provide a more detailed explanation of its construction and interpretation. We hope this makes the figure easier to understand.

“Fig. S1. Taxonomic structure dynamics of large herbivores. Succession of taxonomic modules (M1 to M35) derived from network analysis, plotted against time (in million years, Ma). Each module represents a distinct taxonomic structure based on species and continent-stage (nodes in the network) associations. Different colors and shapes indicate continent-stages across the different continents (America, Europe, Asia, and Africa; see legend). Lines represent module transitions within each continent, illustrating large-scale patterns of taxonomic succession.”

L. 291-293: “The aggregation of localities by continent and stage yielded a total of 78 continent-stage assemblages, which are the analytical basis for the study”. I think that’s totally needed and fair for a study at the temporal and spatial scale, but could you develop a bit the potential limitation of such approach in your method section (or in supplementary)? What could be the impact of such data agglomeration? For example, could different UFFs co-exist on one continent (e.g. SE Asia vs Central Asia) and thus create fake stability by experiencing different trajectories of changes through times?

We used the continent-stage aggregation to correct for potential biases in spatial and temporal sampling. Prior to this aggregation, we carefully checked each occurrence to refine the temporal range, correct locality names, and verify taxonomic assignments. This work was carried out in collaboration with expert taxonomists (coauthors of the study).

To minimize spatial biases, we first aggregated localities by continent. This allowed us to examine macro-scale patterns related to major abiotic changes, without the confounding effects of uneven geographical sampling. We then aggregated the data by chronostratigraphic stages, which vary in temporal range. This approach narrows the temporal range toward the present to address potential bias, especially the tendency to pool more recent data.

The continent-stage aggregation yielded a total of 78 continent-stage assemblages. We chose this approach due to the lack of precise locality coordinates for most of the data, which made it unfeasible to accurately map localities within specific regions or countries (smaller scales). By focusing on a continental scale, we were able to utilize a large number of localities and study their patterns of change in terms of functional structure.

Given our definition of ungulate functional faunas (ungulate continental assemblages with shared distributions of functional types), it is not possible for more than one UFF to co-exist on the same continent (e.g., Southeast Asia vs. Central Asia). Our analysis focuses on the functional structure of large herbivores at a continental scale, so the stability or reassembly observed is at that macro scale, not at the local or regional level. Therefore, we do not believe our results represent "fake" stability; rather, they indicate real stability or reassembly, but at the continental level (UFFs) across the study period.

In response to the reviewer's suggestion, we have included a section in the supplementary materials discussing the potential limitations of the spatio-temporal standardization and how it might influence the interpretation of our results.

I think that the authors could have spend more time to compare known patterns of taxonomic diversity in mammals that took interest in clustering temporal fauna (e.g. chronofauna in both North America and Europe/Asia/Africa, see for example Figueirido, Gibert, Casanovas-Vilar) and look for breaks and tendencies but I understand that Nature format is very constrained. However, I think that the excellent work they did with functional diversity could have

been even better if it would have been better compared to other dimensions of mammalian fossil biodiversity.

As the reviewer pointed out, due to the limited format, we could not discuss all the biotic and abiotic events that took place during the Cenozoic. We focused our discussion on those events that had the greatest impact on functional structure and diversity dynamics, particularly those recorded at continental and global scales. We agree that regional events are of significant interest for the evolution of certain mammalian families. In Blanco et al. 2021 and Nascimento et al. 2024, we addressed some of these regional patterns, particularly in the Iberian Peninsula, due to the regional scale of those analyses. However, the macro-scale approach of our current analysis led us to focus on the broader patterns described in the manuscript.

L. 74-75 : “Since 10 Ma, large herbivore assemblages have experienced remarkable stability”.

One remark: it could be stated here that Asian UFFs are acting slightly differently than the others continental fauna.

Okay, we have slightly rephrased the sentence to incorporate the reviewer's comment:

“Since 10 Ma, large herbivore assemblages have experienced remarkable stability beginning in Africa, followed by America and Europe around 7 Ma, and culminating in the later stability of Asia around 4.5 Ma (Fig. 1A).”

One question: when looking at Fig. 1B, we can see how the average/median point of UFFs in NMDS is moving in the environmental space through time, but I struggle to understand how Fig. 1A and Fig. 1B relate. For example, when looking at the orange/yellow section for Africa (i.e. the last 10 myr), we see strong displacement in the environmental space for the african UFF, but it corresponds to stasis (or a single mode, turquoise-green) in Fig. 1A. In the same way, the early Cenozoic asian UFF relate to the first mode (i.e. red) and last for ~35 myr, but we can see in Fig. 1B this UFF moving a lot through the environmental fauna (i.e. black to red

section of the line). Can you extend the explanation in supplementary data to understand the relationship between Fig. 1A and 1B. Fig. S19 could be used to explain this relationship with more details.

In Figure 1B, we plotted the functional space of all the continent-stage assemblages (represented as dots). The purpose of this figure is to illustrate the different trajectories in terms of functional beta diversity, calculated here as turnover (see Methods), that the continents followed over time, occupying different regions of the functional space.

The relationship between Fig. 1A and 1B lies in the similar trajectories followed by the continents when they formed a joint module, indicating that they explored similar regions of the functional space. Therefore, in Fig. 1B, the key point is the direction of the trajectory movement and the exploration of these shared regions of functional space, rather than the magnitude of change within the space.

To clarify this relationship, we have added text to the Methods section to explain the functional space and its connection to Figs. 1A.

L. 97-98 : “This was primarily driven by the emergence of functional dominance of FTs in larger body size categories in the Americas, in contrast to the mid-sized herbivores that characterize the UFF of Eurasia (Fig. S17)”. Could it be a matter of the relationship between landmass and (paleo)latitudes? The larger southern extent of Asia could explain the survival of mid-sized herbivores and the continuous cooling of Cenozoic could act in very different direction depending on the latitudinal distribution of landmass, isn’t it? The argument developed in the rest of this section (from L.95 to L.119) could be completed by a discussion around landmass/biome (i.e. closed vs open forests) distribution.

We agree with the reviewer that this is an interesting and relevant point. Accordingly, we have added the following text to address this comment L101:

"The geographical extent of landmasses could have significantly influenced the survival dynamics of different faunal groups. Specifically,

the broader southern reach of the Eurasian landmass compared to North America may have provided a more stable environment for mid-sized herbivores. This stability persisted despite the overarching trend of Cenozoic cooling, which likely affected species differently depending on their latitudinal distribution."

The two references associated with EOC transition (29, 40) are associated with very descriptive arguments (L.109-119) when explanatory argument (i.e. mechanism) could/should be invoked here (e.g. Latitude and landmass, Europe being restricted to high latitude and forming an archipelago making climate tracking more difficult for European fauna). On a more general statement, I think that this paper is sometimes a bit too descriptive, more potential mechanisms could be invoked to explain the observed patterns.

We appreciate the reviewer's suggestion to incorporate more mechanistic explanations rather than relying solely on descriptive arguments. To address this, we have expanded our discussion of the EOC transition by explicitly considering the role of latitude, landmass configuration, and geographic constraints on faunal dynamics. Specifically, we discuss how Europe's restricted high-latitude position and archipelagic nature may have limited the ability of its fauna to track climatic shifts compared to other continents. We have added the following text to the main manuscript to strengthen this section in line 126:

"The biogeographic and climatic context of the EOC transition also played a crucial role in shaping the observed functional shifts (29). Europe's restricted high-latitude position and its configuration as an archipelago during this period may have constrained faunal dispersal and climate tracking, making species more vulnerable to extinction as conditions changed. In contrast, larger and more continuous landmasses, such as those in North America and Asia, provided broader climatic gradients and refugia, which facilitated species persistence and functional stability (Fig. 1A). These geographic constraints, coupled with the rapid cooling and heightened seasonality of the EOC transition (29), contribute to explaining the pronounced shifts observed in European faunal assemblages. "

L. 124-125: “The functional coalescence is reflected in the drastic reduction of the functional distance between Africa and Eurasia after 21 Ma”. Very interesting, could you just add the age uncertainties (i.e. the corresponding geological stage length if I correctly understood your methodology) in order to make this statement more subtle. The exact date of the main part of the interchange/faunal mixing between Africa and Eurasia remain debated and your sentence could imply that your study can precisely date this event.

We thank the reviewer for pointing this out. We agree that this is a valuable observation. To refine our statement, we used the break in the segmented regression from the linear model (25.4249 Ma), which marks the start of the reduction in functional distance between Africa and Europe. The reduction between Asia and Africa began later, during the Aquitanian (21.735 Ma). Therefore, we have emphasized the beginning of the reduction in functional distance between Africa and Eurasia. We have added the corresponding geological stage length and refined the ages as follows in line 139:

“The functional coalescence is reflected in the drastic reduction of the functional distance between Africa and Eurasia, beginning around 25 Ma, during the Chantian stage (27.82–23.03 Ma) in Europe, and later followed by the Aquitanian (23.03-20.44 Ma) in Asia (Fig. S3)”

L. 298: “We excluded the species with no information about these traits”. Did you analyse/reanalyse/measure species traits or did you find the 13 traits being already described for the 3012 species included in your study?

Our procedure for assigning traits to species was rigorous and involved multiple sources and expert validation. First, we downloaded all available information from the NOW database, the PBDB, and selected literature. Each species was then individually assessed by taxonomy specialists who are experts in their respective groups and coauthors of this manuscript. Traits were also inferred from bone measurements or extrapolated from characteristics of closely related species when necessary. Thus, these experts confirmed the validity of the trait

assignments. Certain groups, such as Proboscideans, were nearly complete, with most data sourced from a previous work by Cantalapiedra et al. (2021). However, other groups posed challenges due to the scarcity of published information (e.g., Anthracoteriids). Overall, the trait assignment process was a collaborative effort involving 10 specialists over more than 2 years to complete the database.

L. 306: “and assigned a weight of 1/6 to the remaining 12 traits”. Have you tried different weight, in other word, did you perform sensitivity analysis on this parameter ?

We applied different weights to the traits when calculating the distance matrix using the R function `daisy`:

```
daisy(no_na_fdata_art, metric="gower", type =  
list(ordratio = ordratio), weights = weights)
```

This distance matrix was then used to perform a PCOA, followed by k-means clustering to define the functional types. We assigned higher weights to body size and tooth shape because these traits play a significant role in defining the ecological function of species. Without weighting, their effects would be dispersed across the other 12 dental traits.

To balance the combined effect of body size and tooth shape with the remaining 12 dental traits, we used a weight ratio of 1:1/6. This ensures that the influence of the 12 dental traits is roughly equivalent to the combined effect of body size and tooth shape. Altering the weight values would not change the key aspect of this approach, which is that the combined effect of the two sets of traits (i.e., 3 for body size and tooth shape and 3/6 for the remaining 12 traits) remains consistent.

This weighting strategy follows the methodology outlined in Cantalapiedra et al. (2021).

L. 393-397: Thank for the clear interpretation of the Shannon index for FD. Would you say that high Shannon values in your study relate to strong ecosystem stability (high diversity + high redundancy)? The comparison of Shannon FD and Fig. 1 is not trivial.

Thank you for this interesting point. In our study, we observed a peak in the Shannon index around 10 Ma (Fig. 2), which coincides with a transitional phase in both functional diversity and functional structure (Fig. 1 and 2). After this point, we identified a break in the trend of the curve through segmented regression, followed by a sustained drop in functional diversity. This shift coincided with a tipping point in the functional structure, marking the onset of a global reassembly process.

It is true that between 21 and 10 Ma, we observed a continuous increase in functional diversity (Fig. 2), which coincided with a period of functional stability in terms of functional structure (Fig. 1A). However, this sustained increase in functional diversity began earlier during the Cenozoic, when functional reassembly occurred across all continents (Fig. 1A).

Moreover, after the reassembly that began around 10 Ma, the system regained stability, which persisted until the present day, overcoming a continuous decline in functional diversity (Fig. 2).

We acknowledge that the interpretation of Shannon diversity in relation to ecosystem stability is complex. While high functional diversity values are often associated with greater ecosystem resilience and stability, our study suggests that moments of high diversity do not always correspond to long-term ecosystem stability. Instead, the dynamics are influenced by several factors, including the broader context of functional structure. For instance, the period of high diversity observed between 21 and 10 Ma was a time of functional reassembly, which was a process that unfolded across continents (Fig. 1A). This highlights the notion that ecosystem stability is not only a result of high diversity, but also of the resilience and structure of the functional groups within the system. In fact, stability can be context-dependent, influenced by factors such as tectonics, climate change, and other abiotic processes. Therefore, while high functional diversity may suggest a resilient ecosystem, it is not always directly linked to sustained stability, as the functional structure and the system's ability to withstand or recover from disturbances also play crucial roles.

References:

S. Bekeraite, I. Juchnevičiūtė, & A. Spiridonov (2024). Bayesian network analysis reveals the assembly drivers and emergent stability of Eurasian Pleistocene large mammal communities. *Journal of Mammalian Evolution*, 31(4), 1-16.

F. Blanco *et al.*, Punctuated ecological equilibrium in mammal communities over evolutionary time scales. *Science* **372**, 300-303 (2021).

J. Brits, M. Van Rooyen, N. Van Rooyen, Ecological impact of large herbivores on the woody vegetation at selected watering points on the eastern basaltic soils in the Kruger National Park. *African Journal of Ecology* **40**, 53-60 (2002).

J. L. Cantalapiedra *et al.*, The rise and fall of proboscidean ecological diversity. *Nature Ecology & Evolution* **5**, 1266-1272 (2021).

B. Figueirido, C. M. Janis, J. A. Pérez-Claros, M. De Renzi, P. Palmqvist, Cenozoic climate change influences mammalian evolutionary dynamics. *Proceedings of the National Academy of Sciences* **109**, 722-727 (2012).

B. Figueirido, P. Palmqvist, J. A. Pérez-Claros, C. M. Janis, Sixty-six million years along the road of mammalian ecomorphological specialization. *Proceedings of the National Academy of Sciences* **116**, 12698-12703 (2019).

J.C. Nascimento *et al.* (2024). The reorganization of predator–prey networks over 20 million years explains extinction patterns of mammalian carnivores. *Ecology letters*, 27(6), e14448.

F. A. Smith, C. E. Doughty, Y. Malhi, J. C. Svenning, J. Terborgh (2016) Megafauna in the Earth system. (Wiley Online Library), pp 99-108.

C. van der Waal *et al.*, Large herbivores may alter vegetation structure of semi-arid savannas through soil nutrient mediation. *Oecologia* **165**, 1095-1107 (2011).

I. Žliobaitė & M. Fortelius (2022). On calibrating the completeness for the mammalian fossil record. *Paleobiology*, 48(1), 1-11.

Introduction

Lines 48 – 53: I think it would be helpful for the geographic distribution to be stated somewhere in the description of the data here. As a reader, it was the first question that popped in my head. Just adding “global” would be sufficient.

Results and Discussion

Lines 78: Can you clarify why North and South America were analyzed together? Due to the independence of the two faunas, it seems more reasonable that they would be analyzed separately. It seems important to note if their separation alters the overall functional structure. Moreover, you would potentially expect a shift at GABI.

Lines 68 – 73: What defines a functional reconfiguration? And how does that vary from tipping points? You say there is a functional reconfiguration every ~6 Myr, but there are long periods of ecological stasis and there are two periods of reassembly. Could you clarify how those different types of events are defined? I became confused when you said “with a first reconfiguration in UFFs around 40 Ma (line 86)

Line 149: I would replace “recovered” with identified. I had to read it twice because I thought you were suggesting that functional diversity was recovering in those continents.

Lines 197 – 198: If you were looking at functional diversity of North and South America together before GABI, you wouldn't suspect a change in the overall functional pool following the interchange.

Lines 198 – 199: The functional structure has remained the same following all the extinctions and extirpations? If you mean in the most recent interval (encompassing the last 10,000 years), I would say that, not until present day. The way I understand it, is all large ungulates that have existed on the continent during any part of that interval is included in determine the functional structure.

Methods

Lines 297 – 298: It is unclear to me if “no information about these traits” means the species needed to be missing all of the traits or if they are missing a information about a single trait. If species were kept that were missing 1 or multiple traits, could this bias your data and was a sensitivity analysis used to address any possible biases?

Lines 395 – 397 and 404 - 406: a homogenous distribution in functional space does not necessarily represent high redundancy. It is possible to get lower redundancy and an increase in distribution homogeneity across functional space. So I'm not sure I believe high Shannon values always represent ecological redundancy, unless I am misunderstanding something that addresses this in the equation.

Is there any reason to think that geographic distribution of available occurrences could influence your results during any time periods?

Figures

In Fig. 1A African data begins around 30 Ma, in Fig. S2 it begins around 25 Ma and in Fig. 2, African data does not start until 20 Ma. Can you explain why?